# Transformers are Meta-Reinforcement Learners

## Abstract

The transformer architecture and variants presented a remarkable success across many machine learning tasks in recent years. This success is intrinsically related to the capability of handling long sequences and the presence of context-dependent weights from the attention mechanism. We argue that these capabilities suit the central role of a Meta-Reinforcement Learning algorithm. Indeed, a meta-RL agent needs to infer the task from a sequence of trajectories. Furthermore, it requires a fast adaptation strategy to adapt its policy for a new task - which can be achieved using the self-attention mechanism. In this work, we present TrMRL (**Tr**ansformers for **M**eta-**R**einforcement **L**earning), a meta-RL agent that mimics the memory reinstatement mechanism using the transformer architecture. It associates the recent past of working memories to build an episodic memory recursively through the transformer layers. This memory works as a proxy to the current task, and we condition a policy head on it. We conducted experiments in high-dimensional continuous control environments for locomotion and dexterous manipulation. Results show that TrMRL achieves or surpasses state-of-the-art performance, sample efficiency, and out-of-distribution generalization in these environments.

## 1 Introduction

In recent years, the Transformer architecture (Vaswani et al., 2017) achieved exceptional performance on many machine learning applications, especially for text (Devlin et al., 2019; Raffel et al., 2020) and image processing (Dosovitskiy et al., 2021b; Caron et al., 2021; Yuan et al., 2021). This intrinsically relates to its few-shot learning nature Brown et al. (2020a): the attention weights work as context-dependent parameters, inducing better generalization. Furthermore, this architecture parallelizes token processing by design. This capability avoids the vanishing gradients problem, very common for recurrent models. As a result, they can handle longer sequences more efficiently.

This work argues that these two capabilities are essential for a Meta-Reinforcement Learning (meta-RL) agent. We propose TrMRL (**Tr**ansformers for **M**eta-**R**einforcement **L**earning), a memory-based meta-Reinforcement Learner which uses the transformer architecture to formulate the learning process. It works as a memory reinstatement mechanism (Rovee-Collier, 2012) during learning, associating recent working memories to create an episodic memory which is used to contextualize the policy.

Figure 1 illustrates the process. We formulated each task as a distribution over working memories. TrMRL associates these memories using self-attention blocks to create a task representation in each head. These task representations are combined in the position-wise MLP to create an episodic output (which we identify as episodic memory). We recursively apply this procedure through layers to refine the episodic memory. In the end, we select the memory associated with the current timestep and feed it into the policy head.

Nonetheless, transformer optimization is often unstable, especially in the RL setting. Past attempts either fail to stabilize (Mishra et al., 2018) or required architectural additions (Parisotto et al., 2019) or restrictions on the observations space (Loynd et al., 2020). We argue that this challenge can be mitigated through a proper weight initialization scheme. For this matter, we applied T-Fixup initialization (Huang et al., 2020).

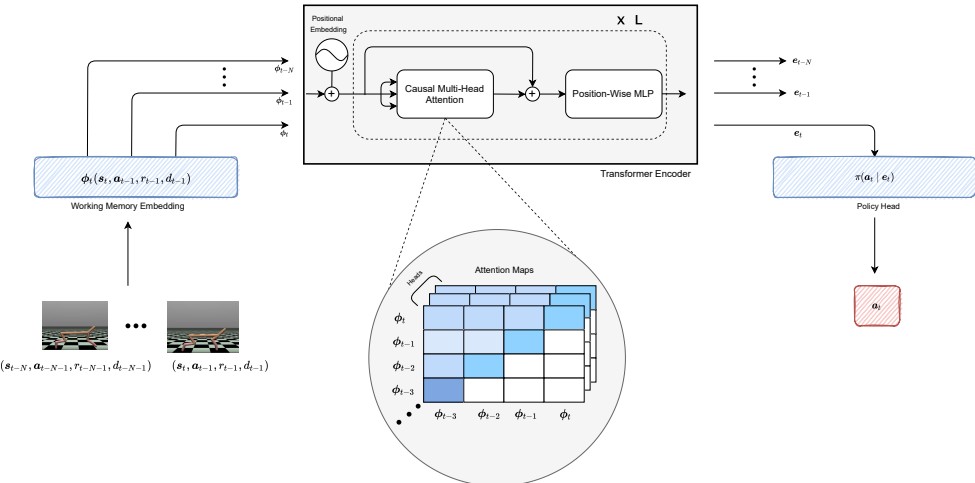

Figure 1: Illustration of the TrMRL agent. At each timestep, it associates the recent past of working memories to build an episodic memory through transformer layers recursively. We argue that the self-attention works as a fast adaptation strategy since it provides context-dependent parameters.

We conducted a series of experiments to evaluate meta-training, fast adaptation, and out-of-distribution generalization in continuous control environments for locomotion and robotic manipulation. Results show that TrMRL consistently achieves or surpasses the current state-of-the-art meta-RL agents in performance and sample efficiency. It presents online adaptation, requiring as few as 20 timesteps to identify and achieve the desired performance on test tasks. We also conducted an ablation study to show the effectiveness of the T-Fixup initialization, and the sensibility to network depth, sequence size, and the number of attention heads.

## 2 RELATED WORK

**Meta-Learning** is an established Machine Learning (ML) principle to learn inductive biases from the distribution of tasks to produce a data-efficient learning system (Bengio et al., 1991; Schmidhuber et al., 1996; Thrun & Pratt, 1998). This principle spanned in a variety of methods in recent years, learning different components of an ML system, such as the optimizer (Andrychowicz et al., 2016; Li & Malik, 2016; Chen et al., 2017), neural architectures (Hutter et al., 2019; Zoph & Le, 2017), metric spaces (Vinyals et al., 2016), weight initializations (Finn et al., 2017; Nichol et al., 2018; Finn et al., 2018), and conditional distributions (Zintgraf et al., 2019; Melo et al., 2019). Another branch of methods learns the entire system using memory-based architectures (Ortega et al., 2019; Wang et al., 2017; Duan et al., 2016; Ritter et al., 2018a) or generating update rules by discovery (Oh et al., 2020) or evolution (Co-Reyes et al., 2021).

**Memory-Based Meta-Learning** is the particular class of methods where we focus on in this work. In this context, Wang et al. (2017); Duan et al. (2016) concurrently proposed the $RL^2$ framework, which formulates the learning process as a Recurrent Neural Network (RNN) where the hidden state works as the memory mechanism. Given the recent rise of attention-based architectures, one natural idea is to use it as a replacement for RNNs. Mishra et al. (2018) proposed an architecture composed of causal convolutions (to aggregate information from past experience) and soft-attention (to pinpoint specific pieces of information). In contrast, our work applies causal, multi-head self-attention by stabilizing the complete transformer architecture with an arbitrarily large context window. Finally, Ritter et al. (2021) also applied multi-head self-attention for rapid task solving for RL environments. However, in a different dynamic: their work applied the attention mechanism iteratively in a pre-defined episodic memory, while ours applies it recursively through transformer layers to build an episodic memory from the association of recent working memories.

Our work has intersections with Cognitive Neuroscience research on memory for learning systems (Hoskin et al., 2018; Rovee-Collier, 2012; Wang et al., 2018). In this context, Ritter et al. (2018c)

extended the RL$^2$ framework incorporating a differentiable neural dictionary as the inductive bias for episodic memory recall. In the same line, Ritter et al. (2018b) also extended RL$^2$ but integrating a different episodic memory system inspired by the reinstatement mechanism. In our work, we also mimic reinstatement to retrieve episodic memories from working memories but using self-attention. Lastly, Fortunato et al. (2019) studied the association between working and episodic memories for RL agents, specifically for memory tasks, proposing separated inductive biases for these memories based on LSTMs and auxiliary unsupervised losses. In contrast, our work studies this association for the Meta-RL problem, using memory as a task proxy implemented by the transformer architecture.

**Meta-Reinforcement Learning** is a branch of Meta-Learning for RL agents. Some of the algorithms described in past paragraphs extend to the Meta-RL setting by design (Finn et al., 2017; Mishra et al., 2018; Wang et al., 2017; Duan et al., 2016). Others were explicitly designed for RL and often aimed to create a task representation to condition the policy. PEARL (Rakelly et al., 2019) is an off-policy method that learns a latent representation of the task and explores via posterior sampling. MAESN (Gupta et al., 2018) also creates task variables but optimizes them with on-policy gradient descent and explores by sampling from the prior. MQL (Fakoor et al., 2020) is also an off-policy method, but it uses a deterministic context that is not permutation invariant implemented by an RNN. Lastly, VariBAD Zintgraf et al. (2020) formulates the problem as a Bayes-Adaptive MDP and extends the RL$^2$ framework by incorporating a stochastic latent representation of the task trained with a VAE objective. Our work contrasts all the previous methods in this task representation: we condition the policy in the episodic memory generated by the transformer architecture from the association of past working memories. We show that this episodic memory works as a *proxy* to the task representation.

**Transformers for RL.** The application of the transformer architecture in the RL setting is still an open challenge. Mishra et al. (2018) tried to apply this architecture for simple bandit tasks and tabular MDPs and reported unstable train and random performance. Parisotto et al. (2019) then proposed some architectural changes in the vanilla transformer, reordering layer normalization modules and replacing residual connections with expressing gating mechanisms, improving state-of-the-art performance for a set of memory environments. Loynd et al. (2020) also studied how transformer-based models can improve the performance of sequential decision-making agents. It stabilized the architecture using factored observations and an intense hyperparameter tuning procedure, resulting in improved sample efficiency. In contrast to these methods, our work stabilizes the transformer model by improving optimization through a better weight initialization. In this way, we could use the vanilla transformer without architectural additions or imposing restrictions on the observations.

Finally, recent work studied how to replace RL algorithms with transformer-based language models (Janner et al., 2021; Chen et al., 2021). Using a supervised prediction loss in the offline RL setting, they modeled the agent as a sequence problem. Our work, on the other hand, considers the standard RL formulation in the meta-RL setting.

## 3 PRELIMINARIES

We define a Markov decision process (MDP) by a tuple $\mathcal{M} = (\mathcal{S}, \mathcal{A}, \mathcal{P}, \mathcal{R}, \mathcal{P}_0, \gamma, H)$, where $\mathcal{S}$ is a state space, $\mathcal{A}$ is an action space, $\mathcal{P} : \mathcal{S} \times \mathcal{A} \times \mathcal{S} \to [0, \infty)$ is a transition dynamics, $\mathcal{R} : \mathcal{S} \times \mathcal{A} \to [-R_{max}, R_{max}]$ is a bounded reward function, $\mathcal{P}_0 : \mathcal{S} \to [0, \infty)$ is an initial state distribution, $\gamma \in [0, 1]$ is a discount factor, and $H$ is the horizon. The standard RL objective is to maximize the cumulative reward, i.e., $\max \mathbb{E}[\sum_{t=0}^{T} \gamma^t \mathcal{R}(s_t, a_t)]$, with $a_t \sim \pi_{\boldsymbol{\theta}}(a_t \mid s_t)$ and $s_t \sim \mathcal{P}(s_t \mid s_{t-1}, a_{t-1})$, where $\pi_{\boldsymbol{\theta}} : \mathcal{S} \times \mathcal{A} \to [0, \infty)$ is a policy parameterized by $\boldsymbol{\theta}$.

### 3.1 PROBLEM SETUP: META-REINFORCEMENT LEARNING

In the meta-RL setting, we define $p(\mathcal{M}) : \mathcal{M} \to [0, \infty)$ a distribution over a set of MDPs $\mathcal{M}$. During meta-training, we sample $\mathcal{M}_i \sim p(\mathcal{M})$ from this distribution, where $\mathcal{M}_i = (\mathcal{S}, \mathcal{A}, \mathcal{P}_i, \mathcal{R}_i, \mathcal{P}_{0,i}, \gamma, H)$. Therefore, the tasks[1] share a similar structure in this setting, but reward function and transition dynamics vary. The goal is to learn a policy that, during meta-testing, can adapt to a new task sampled from the same distribution $p(\mathcal{M})$. In this context, adaptation means

---

[1]We use the terms task and MDP interchangeably.

maximizing the reward under the task in the most efficient way. To achieve this, the meta-RL agent should learn the prior knowledge shared across the distribution of tasks. Simultaneously, it should learn how to differentiate and identify these tasks using only a few episodes.

## 3.2 TRANSFORMER ARCHITECTURE

The transformer architecture (Vaswani et al., 2017) was first proposed as an encoder-decoder architecture for neural machine translation. Since then, many variants have emerged, proposing simplifications or architectural changes across many ML problems (Dosovitskiy et al., 2021a; Brown et al., 2020b; Parisotto et al., 2019). Here, we describe the encoder architecture as it composes our memory-based meta-learner.

The transformer encoder is a stack of multiple equivalent layers. There are two main components in each layer: a multi-head self-attention block, followed by a position-wise feed-forward network. Each component contains a residual connection (He et al., 2015) around them, followed by layer normalization (Ba et al., 2016). The multi-head self-attention (MHSA) block computes the self-attention operation across many different heads, whose outputs are concatenated to serve as input to a linear projection module, as in Equation 1:

$$\text{MHSA}(K, Q, V) = \text{Concat}(h_1, h_2, \ldots, h_\omega)W_o,$$
$$h_i = \text{softmax}(\frac{QK^T}{\sqrt{d}} \cdot M)V, \tag{1}$$

where $K, Q, V$ are the keys, queries, and values for the sequence input, respectively. Additionally, $d$ represents the dimension size of keys and queries representation and $\omega$ the number of attention heads. $M$ represents the attention masking operation. $W_o$ represents a linear projection operation.

The position-wise feed-forward block is a 2-layer dense network with a ReLU activation between these layers. All positions in the sequence input share the parameters of this network, equivalently to a $1 \times 1$ temporal convolution over every step in the sequence. Finally, we describe the positional encoding. It injects the relative position information among the elements in the sequence input since the transformer architecture fully parallelizes the input processing. The standard positional encoding is a sinusoidal function added to the sequence input (Vaswani et al., 2017).

## 3.3 T-FIXUP INITIALIZATION

The training of transformer models is notoriously difficult, especially in the RL setting (Parisotto et al., 2019). Indeed, gradient optimization with attention layers often requires complex learning rate warmup schedules to prevent divergence (Huang et al., 2020). Recent work suggests two main reasons for this requirement. First, the Adam optimizer (Kingma & Ba, 2017) presents high variance in the inverse second moment for initial updates, proportional to a divergent integral (Liu et al., 2020). It leads to problematic updates and significantly affects optimization. Second, the backpropagation through layer normalization can also destabilize optimization because the associated error depends on the magnitude of the input (Xiong et al., 2020).

Given these challenges, Huang et al. (2020) proposed a weight initialization scheme (T-Fixup) to eliminate the need for learning rate warmup and layer normalization. This is particularly important to the RL setting once current RL algorithms are very sensitive to the learning rate for learning and exploration.

T-Fixup appropriately bounds the original Adam update to make variance finite and reduce instability, regardless of model depth. We refer to Huang et al. (2020) for the mathematical derivation. We apply the T-Fixup for the transformer encoder as follows:

- Apply Xavier initialization (Glorot & Bengio, 2010) for all parameters excluding input embeddings. Use Gaussian initialization $\mathcal{N}(0, d^{-\frac{1}{2}})$, for input embeddings, where $d$ is the embedding dimension;
- Scale the linear projection matrices in each encoder attention block and position-wise feed-forward block by $0.67N^{-\frac{1}{4}}$.

## 4 TRANSFORMERS ARE META-REINFORCEMENT LEARNERS

In this work, we argue that two critical capabilities of transformers compose the central role of a Meta-Reinforcement Learner. First, transformers can handle long sequences and reason over long-term dependencies, which is essential to the meta-RL agent to identify the MDP from a sequence of trajectories. Second, transformers present context-dependent weights from self-attention. This mechanism serves as a fast adaptation strategy and provides necessary adaptability to the meta-RL agent for new tasks.

### 4.1 TASK REPRESENTATION

We represent a working memory at the timestep $t$ as a parameterized function $\phi_t(\boldsymbol{s}_t, \boldsymbol{a}_t, r_t, \eta_t)$, where $\boldsymbol{s}_t$ is the MDP state, $\boldsymbol{a}_t \sim \pi(\boldsymbol{a}_t \mid \boldsymbol{s}_t)$ is an action, $r_t \sim \mathcal{R}(\boldsymbol{s}_t, \boldsymbol{a}_t)$ is the reward, and $\eta_t$ is a boolean flag to identify whether this is a terminal state. Our first hypothesis is that we can define a task $\mathcal{T}$ as a distribution over working memories, as in Equation 2:

$$\mathcal{T}(\phi) : \Phi \to [0, \infty), \qquad (2)$$

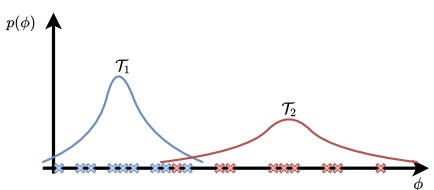

Figure 2: The illustration of two tasks ($\mathcal{T}_1$ and $\mathcal{T}_2$) as distributions over working memories. The intersection of both densities represents the ambiguity between $\mathcal{T}_1$ and $\mathcal{T}_2$.

where $\Phi$ is the working memory embedding space. In this context, one goal of a meta-RL agent is to learn $\phi$ to make a distinction among the tasks in the embedding space $\Phi$. Furthermore, the learned embedding space should also approximate the distributions of similar tasks so that they can share knowledge. Figure 2 illustrates this concept for a one-dimensional representation.

We aim to find a representation for the task given its distribution to contextualize our policy. Intuitively, we can represent each task as a linear combination of working memories sampled by the policy interacting with it:

$$\mu_{\mathcal{T}} = \sum_{t=0}^{N} \alpha_t \cdot \mathcal{W}(\phi_t(\boldsymbol{s}_t, \boldsymbol{a}_t, r_t, \eta_t)),$$

$$\text{with} \sum_{t=0}^{N} \alpha_t = 1 \qquad (3)$$

where $N$ represents the length of a segment of sampled trajectories during the policy and task interaction. $\mathcal{W}$ represents an arbitrary linear transformation. Furthermore, $\alpha_t$ is a coefficient to compute how relevant a particular working memory $t$ is to the task representation, given the set of sampled working memories. Next, we show how the self-attention computes these coefficients, which we use to output an episodic memory from the transformer architecture.

### 4.2 SELF-ATTENTION AS A FAST ADAPTATION STRATEGY

In this work, our central argument is that self-attention works as a fast adaptation strategy. The context-dependent weights dynamically compute the working memories coefficients to implement Equation 3. We now derive *how* we compute these coefficients. Figure 3 illustrates this mechanism.

Let us define $\phi_t^k$, $\phi_t^q$, and $\phi_t^v$ as a representation[2] of the working memory at timestep $t$ in the keys, queries, and values spaces, respectively. The dimension of the queries and keys spaces is $d$. We aim to compute the attention operation in Equation 1 for a sequence of $T$ timesteps, resulting in Equation 4:

---

[2] we slightly abuse the notation by omitting the function arguments – $\boldsymbol{s}_t$, $\boldsymbol{a}_t$, $r_t$, and $\eta_t$ – for the sake of conciseness.

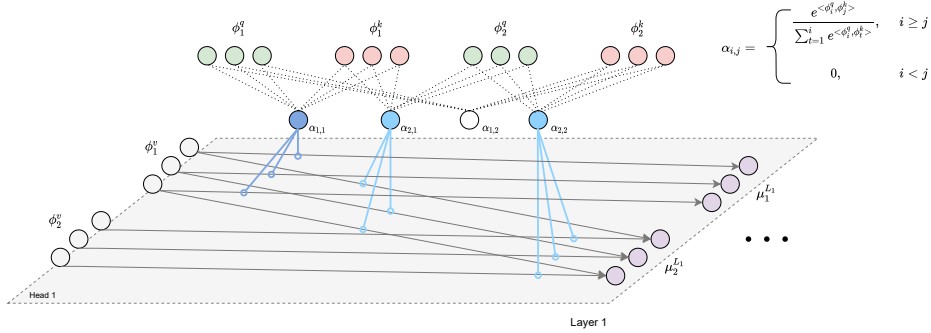

Figure 3: Illustration of causal self-attention as a fast adaptation strategy. In this simplified scenario (2 working memories), the attention weights $\alpha_{i,j}$ drives the association between the current working memory and the past ones to compute a task representation $\mu_t$. Self-attention computes this association by relative similarity.

$$\text{softmax}(\frac{QK^T}{\sqrt{d}} \cdot M)V = \frac{1}{\sqrt{d}} \begin{bmatrix} \alpha_{1,1} & \alpha_{1,2} & \cdots \\ \vdots & \ddots & \\ \alpha_{T,1} & & \alpha_{T,T} \end{bmatrix} \begin{bmatrix} \phi_1^v \\ \vdots \\ \phi_T^v \end{bmatrix} = \begin{bmatrix} \mu_1 \\ \vdots \\ \mu_T \end{bmatrix},$$

$$\text{where} \begin{cases} \alpha_{i,j} = \frac{\exp \langle \phi_i^q, \phi_j^k \rangle}{\sum_{n=1}^i \exp \langle \phi_1^q, \phi_n^k \rangle} & \text{if } i \leq j \\ 0 & \text{otherwise.} \end{cases} \tag{4}$$

where $\langle a_i, b_j \rangle = \sum_{n=0}^d a_{i,n} \cdot b_{j,n}$ is the dot product between the working memories $a_i$ and $b_j$. Therefore, for a particular timestep $t$, the self-attention output is:

$$\mu_t = \frac{1}{\sqrt{d}} \cdot \frac{\phi_1^v \cdot \exp \langle \phi_t^q, \phi_1^k \rangle + \cdots + \phi_t^v \cdot \exp \langle \phi_t^q, \phi_t^k \rangle}{\sum_{n=1}^i \exp \langle \phi_1^q, \phi_n^k \rangle}$$

$$= \frac{1}{\sqrt{d}} \sum_{n=1}^t \alpha_{t,n} W_v(\phi_t). \tag{5}$$

Equation 5 shows that the self-attention mechanism implements the task representation in Equation 3 by associating past working memories *given that* the current one is $\phi_t$. It computes this association with *relative similarity* through the dot product normalized by the softmax operation. This inductive bias helps the working memory representation learning to approximate the density of the task distribution $\mathcal{T}(\phi)$.

### 4.3 TRANSFORMERS AND MEMORY REINSTATEMENT

We now argue that the transformer model implements a memory reinstatement mechanism for episodic memory retrieval. An episodic memory system is a long-lasting memory that allows an agent to recall and re-experience personal events (Tulving, 2002). It complements the working memory system, which is active and relevant for short periods (Baddeley, 2010) and works as a buffer for episodic memory retrieval (Zilli & Hasselmo, 2008). Adopting this memory interaction model, we model an episodic memory as a transformation over a collection of past memories. More concretely, we consider that a transformer layer implements this transformation:

$$e_t^l = f(e_0^{l-1}, \ldots, e_t^{l-1}), \tag{6}$$

where $e_t^l$ represents the episodic memory retrieved from the last layer $l$ for the timestamp $t$ and $f$ represents the transformer layer. Equation 6 provides a recursive definition, and $e_t^0$ (the base case)

corresponds to the working memories $\phi_t$ In this way, the transformer architecture recursively refines the episodic memory interacting memories retrieved from the past layer. We show the pseudocode for this process in Algorithm 1. This refinement is guaranteed by a crucial property of the self-attention mechanism: it computes a consensus representation across the input memories associated to the sub-trajectory, as stated by Theorem 1 (Proof in Appendix E). Here, we define consensus representation as the memory representation that is closest on average to all likely representations (Kumar & Byrne, 2004), i.e., minimizes the Bayes risk considering the set of episodic memories.

**Theorem 1.** *Let $\mathcal{S}^l = (e_0^l, \ldots, e_N^l) \sim p(e|\mathcal{S}^l, \boldsymbol{\theta}_l)$ be a set of normalized episodic memory representations sampled from the posterior distribution $p(e|\mathcal{S}^l, \boldsymbol{\theta}_l)$ induced by the transformer layer $l$, parameterized by $\boldsymbol{\theta}_l$. Let $K$, $Q$, $V$ be the Key, Query, and Value vector spaces in the self-attention mechanism. Then, the self-attention in the layer $l+1$ computes a consensus representation $e_N^{l+1} = \frac{\sum_{t=1}^N e_t^{l,V} \cdot \exp \langle e_t^{l,Q}, e_i^{l,K} \rangle}{\sum_{t=1}^N \exp \langle e_t^{l,Q}, e_i^{l,K} \rangle}$ whose associated Bayes risk (in terms of negative cosine similarity) lower bounds the Minimum Bayes Risk (MBR) predicted from the set of candidate samples $\mathcal{S}^l$ projected onto the $V$ space.*

Lastly, we condition the policy head in the episodic memory from the current timestep to sample an action. This complete process resembles a memory reinstatement operation: a reminder procedure that reintroduces past elements in advance of a long-term retention test (Rovee-Collier, 2012). In our context, this "long-term retention test" identifies the task and acts accordingly to maximize rewards.

## 5 EXPERIMENTS AND ANALYSIS

In this section, we present an empirical validation of our method, comparing it with the current state-of-the-art methods. We considered high-dimensional, continuous control tasks for locomotion (MuJoCo[3]) and dexterous manipulation (MetaWorld). We describe them in Appendix A. For reproducibility (source code and hyperparameters), we refer to the released source code[4].

### 5.1 EXPERIMENTAL SETUP

**Meta-Training.** During meta-training, we repeatedly sampled a batch of tasks to collect experience with the goal of learning to learn. For each task, we ran a sequence of $E$ episodes. During the interaction, the agent conducted exploration with a gaussian policy. During optimization, we concatenate these episodes to form a single trajectory and we maximize the discounted cumulative reward of this trajectory. This is equivalent to the training setup for other on-policy meta-RL algorithms (Duan et al., 2016; Zintgraf et al., 2020). For these experiments, we considered $E = 2$. We performed this training via Proximal Policy Optimization (PPO) (Schulman et al., 2017), and the data batches mixed different tasks. Therefore, we present here an on-policy version of the TrMRL algorithm. To stabilize transformer training, we used the T-Fixup as a weight initialization scheme.

**Meta-Testing.** During meta-testing, we sampled new tasks. These are different from the tasks in meta-training, but they come from the same distribution, except during Out-of-Distribution (OOD) evaluation. For TrMRL, in this stage, we froze all network parameters. For each task, we ran few episodes, performing the adaptation strategy. The goal is to identify the current MDP and maximize the cumulative reward across the episodes.

**Memory Write Logic.** At each timestep, we fed the network with the sequence of working memories. This process works as follows: at the beginning of an episode (when the memory sequence is empty), we start writing the first positions of the sequence until we fill all the slots. Then, for each new memory, we removed the oldest memory in the sequence (in the "back" of this "queue") and added the most recent one (in the "front").

**Comparison Methods.** For comparison, we evaluated three different state-of-the-art meta-RL algorithms: $RL^2$ Duan et al. (2016), optimized using PPO Schulman et al. (2017); PEARL (Rakelly et al., 2019); and MAML (Finn et al., 2017), whose outer-loop used TRPO (Schulman et al., 2015).

---

[3]We highlight that both MuJoCo (Locomotion Tasks) and MetaWorld are built on the MuJoCo physics engine. We identify the set of locomotion tasks as solely for "MuJoCo" to ensure simpler and concise writing during analysis of the results.

[4]Omitted due to double-blind review. See supplementary material.

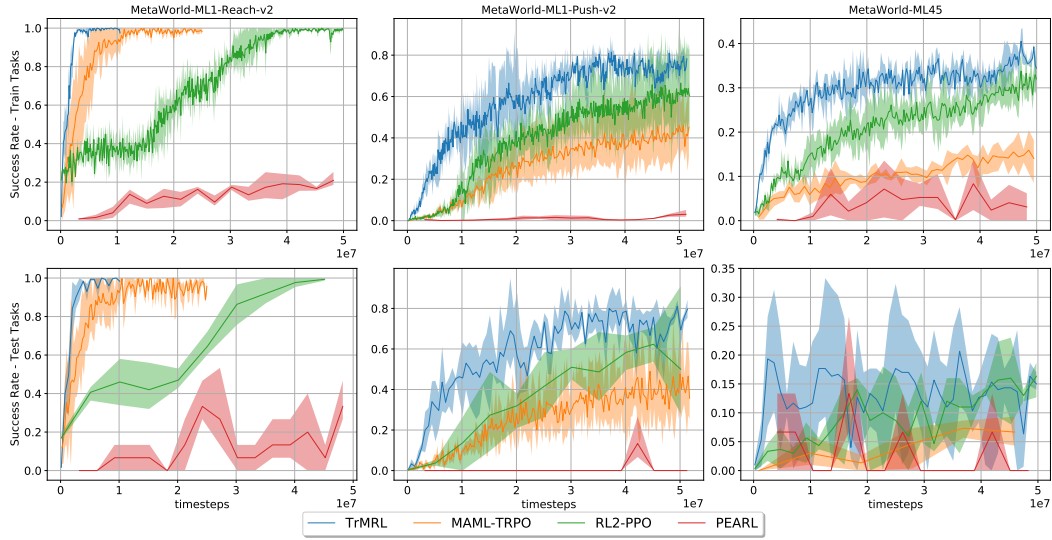

Figure 4: Meta-Training results for MetaWorld benchmarks. The plots on top represents performance on training tasks, while the plots on bottom represents in the test tasks.

## 5.2 RESULTS AND ANALYSIS

We compared TrMRL with baseline methods in terms of meta-training, episode adaptation, and OOD performance. We also present the latent visualization for TrMRL working memories and ablation studies. All the curves presented are averaged across three random seeds, with 95% bootstrapped confidence intervals.

**Meta-Training Evaluation**. Figure 4 shows the meta-training results for all the methods in the MetaWorld environments. All subplots show the task success rate over the training timesteps. The plots on top represent performance on training tasks, while the plots on the bottom represent on the test tasks. TrMRL outperformed all baseline methods. In the "Reach-v2", TrMRL, RL$^2$, and MAML reached the perfect success rate, but TrMRL was more sample efficient. For more complex scenarios, such as "Push-v2" and ML45, TrMRL still performs consistently better. Nevertheless, we also present that all the presented methods performed poorly on ML45 for test tasks, highlighting a big improvement room. We hypothesize that this is due to the lack of a meta-exploration strategy and an inductive bias to improve knowledge share among different tasks. For MuJoCo locomotion tasks, we refer to Appendix B.

**Fast Adaptation Evaluation**. A critical skill for meta-RL agents is the capability of adapting to new tasks given a few episodes. We evaluate this by running meta-testing on 20 test tasks over 6 sequential episodes. Each agent runs its adaptation strategy to identify the task and maximize the reward across episodes. Figure 6 presents the results for the locomotion tasks. For AntDir and HalfCheetaVel, TrMRL outperformed all methods. For HalfCheetahDir, TrMRL started with better performance, but PEARL outperformed after running its adaptation.

We highlight that TrMRL presented high performance since the first episode. In fact, it only requires a few timesteps to achieve high performance in test tasks. In the HalfCheetahVel, for example, it only requires around 20 timesteps to achieve the best performance (Figure 5). Therefore, it presents a nice

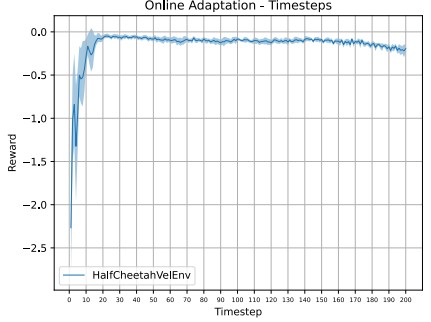

Figure 5: TrMRL's adaptation for HalfCheetahVel environment.

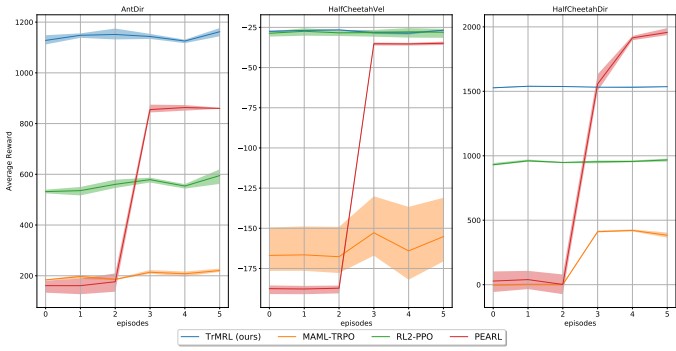

Figure 6: Fast adaptation results on MuJoCo locomotion tasks. Each curve represents the average performance over 20 test tasks. TrMRL presented high performance since the first episode due to the online adaptation nature from attention weights.

property for online adaptation. This is because the self-attention mechanism is lightweight and only requires a few working memories to achieve good performance. Hence, we can run it efficiently at each timestep. Other methods, such as PEARL and MAML, do not present such property, and they need a few episodes before executing adaptation efficiently.

**OOD Evaluation**. Another critical scenario is how the fast adaptation strategies perform for out-of-distribution tasks. For this case, we change the HalfCheetahVel environment to sample OOD tasks during the meta-test. In the standard setting, both training and testing target velocities are sampled from a uniform distribution in the interval $[0.0, 3.0]$. In the OOD setting, we sampled 20 tasks in the interval $[3.0, 4.0]$ and assessed adaptation throughout the episodes. Figure 7 presents the results. TrMRL surpasses all the baselines methods with a good margin, suggesting that the context-dependent weights learned a robust adaptation strategy, while other methods memorized some aspects of the standard distribution of tasks. We especially highlight PEARL, which achieved the best performance among the methods but performed poorly in this setting, suggesting that it does not generate useful latent representations for OOD tasks.

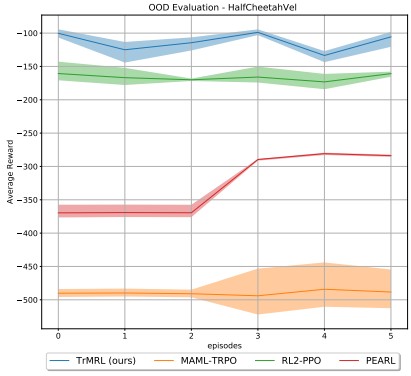

Figure 7: OOD Evaluation in HalfCheetahVel environment.

## 6 CONCLUSION AND FUTURE WORK

In this work, we presented TrMRL, a memory-based meta-RL algorithm built upon a transformer, where the multi-head self-attention mechanism works as a fast adaptation strategy. We designed this network to resemble a memory reinstatement mechanism, associating past working memories to dynamically represent a task and recursively build an episode memory through layers.

TrMRL demonstrated a valuable capability of learning from reward signals. On the other side, recent Language Models presented substantial improvements by designing self-supervised tasks (Devlin et al., 2019; Raffel et al., 2020) or even automating their generation (Shin et al., 2020). As future work, we aim to investigate how to enable these forms of self-supervision to leverage off-policy data collected during the training and further improve sample efficiency in transformers for the meta-RL scenario.

## 7 REPRODUCIBILITY STATEMENT

**Code Release**. To ensure the reproducibility of our research, we released all the source code associated with our models and experimental pipeline. We refer to the supplementary material of this submission. It also includes the hyperparameters and the scripts to execute all the scenarios presented in this paper.

**Baselines Reproducibility**. We strictly reproduced the results from prior work implementations for baselines, and we provide their open-source repositories for reference.

**Proof of Theoretical Results and Pseudocode**. We provide a detailed proof of Theorem 1 in Appendix E, containing all assumptions considered. We also provide pseudocode from TrMRL's agent to improve clarity on the proposed method.

**Availability of all simulation environments**. All environments used in this work are freely available and open-source.

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

# A    METAL-RL ENVIRONMENTS DESCRIPTION

In this Appendix, we detail the environments considered in this work.

## A.1    MUJOCO – LOCOMOTION TASKS

This benchmark is a set of locomotion tasks on the MuJoCo (Todorov et al., 2012) environment. It comprises different bodies, and each environment provides different tasks with different learning goals. These locomotion tasks are previously introduced by Finn et al. (2017) and Rothfuss et al. (2018). We considered 3 different environments.

- **AntDir**: This environment has an ant body, and the goal is to move forward or backward. Hence, it presents these 2 tasks.
- **HalfCheetahDir**: This environment has a half cheetah body, and the goal is to move forward or backward. Hence, it presents these 2 tasks.
- **HalfCheetalVel**: This environment also has a half cheetah body, and the goal is to achieve a target velocity running forward. This target velocity comes from a continuous uniform distribution.

These locomotion task families require adaptation across reward functions.

## A.2    METAWORLD

The MetaWorld (Yu et al., 2021) benchmark contains a diverse set of manipulation tasks designed for multi-task RL and meta-RL settings. MetaWorld presents a variety of evaluation modes. Here, we describe the two modes used in this work. For more detailed description of the benchmark, we refer to Yu et al. (2021).

- **ML1**: This scenario considers a single robotic manipulation task but varies the goal. The meta-training "tasks" corresponds to 50 random initial object and goal positions, and meta-testing on 50 heldout positions.
- **ML45**: With the objective of testing generalization to new manipulation tasks, the benchmark provides 45 training tasks and holds out 5 meta-testing tasks.

These robotic manipulation task families require adaptation across reward functions and dynamics.

# B  MUJOCO: META-TRAINING EVALUATION

In this Appendix, we supplement the meta-training evaluation with the results on MuJoCo locomotion tasks. Figure 8 shows the average return over train tasks (on top) and test tasks (on bottom) for AntDir, HalfCheetahVel, and HalfCheetahDir, respectively. While PEARL failed to explore and learn in robotic manipulation tasks, it presented better results on locomotion tasks, especially in sample efficiency. This is because of its off-policy nature: it efficiently reuses trajectories sampled from previous policy versions, reducing the number of training steps needed. TrMRL achieved the same test task performance in AntDir and HalfCheetahVel as PEARL, also keeping the metric stable over the training. When comparing with other on-policy methods, TrMRL significantly improved performance and sample efficiency.

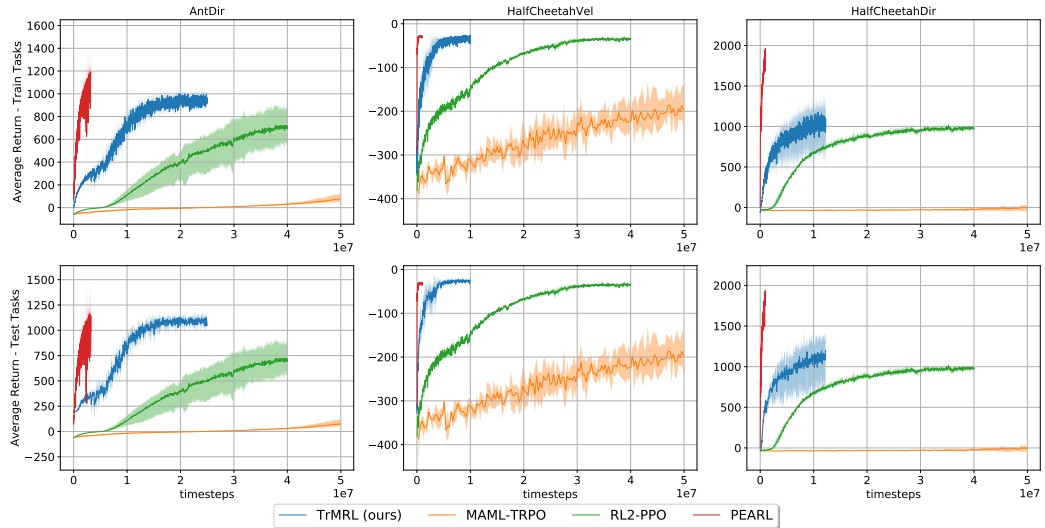

Figure 8: Meta-Training results for MuJoCo locomotion benchmarks. The plots on top represent performance on training tasks, while the plots on bottom represent the test tasks.

## C WORKING MEMORIES LATENT VISUALIZATION

Figure 9 presents a 3-D view of the working memories from the HalfCheetahVel environment. We sampled some tasks (target velocities) and collected working memories during the meta-test setting. We observe that this embedding space learns a representation of each MDP as a distribution over the working memories, as suggested in Section 4. In this visualization, we can draw planes that approximately distinguish these tasks. Working memories that cross this boundary represent the ambiguity between two tasks. Furthermore, this representation also learns the similarity of tasks: for example, the cluster of working memories for target velocity $v = 1.0$ is between the clusters for $v = 0.5$ and $v = 1.5$. This property induces knowledge sharing among all the tasks, which suggests the sample efficiency behind TrMRL meta-training.

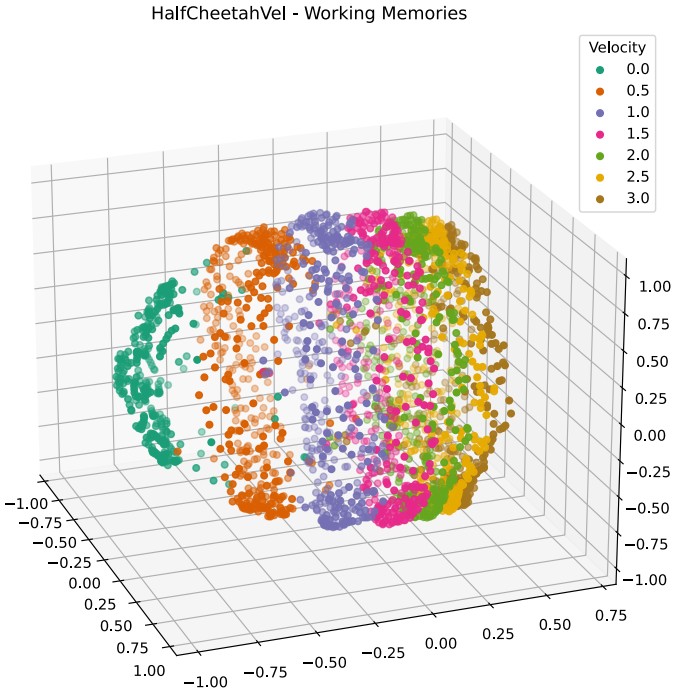

Figure 9: 3-D Latent visualization of the working memories for the HalfCheetahVel environment. We plotted the 3 most relevant components from PCA. TrMRL learns a representation of each MDP as a distribution over the working memories. This representation distinguishes the tasks and approximates similar tasks, which helps knowledge sharing among them.

# D   ABLATION STUDY

In this section, we present an ablation study regarding the main components of TrMRL to identify how they affect the performance of the learned agents. For all the scenarios, we considered one environment for each benchmark to represent both locomotion (HalfCheetahVel) and dexterous manipulation (MetaWorld-ML1-Reach-v2). We evaluated the meta-training phase so that we could analyze both sample efficiency and asymptotic performance.

## D.1   T-FIXUP

In this work, we employed T-Fixup to address the instability from the early stages of transformer training, given the reasons described in Section 3.3. In RL, the early stages of training are also the moment when the learning policies are more exploratory to cover the state and action spaces better and discover rewards, preventing the convergence to sub-optimal policies. Hence, it is crucial for RL that the transformer policy learns appropriately since the beginning to drive exploration.

This section evaluated how T-Fixup becomes essential for environments where the learned behaviors must guide exploration to prevent poor policies. For this, we present T-Fixup ablation (Figure 10) for two settings: MetaWorld-ML1-Reach-v2 and HalfCheetahVel. For the reach environment, we compute the reward distribution using the distance between the gripper and the target location. Hence, it is always a dense and informative signal: even a random policy can easily explore the environment, and T-Fixup does not interfere with the learning curve. On the other side, HalfCheetahVel requires a functional locomotion gate to drive exploration; otherwise, it can get stuck with low rewards (e.g., cheetah is exploring while fallen). In this scenario, T-Fixup becomes crucial to prevent unstable learning updates that could collapse the learning policy to poor behaviors.

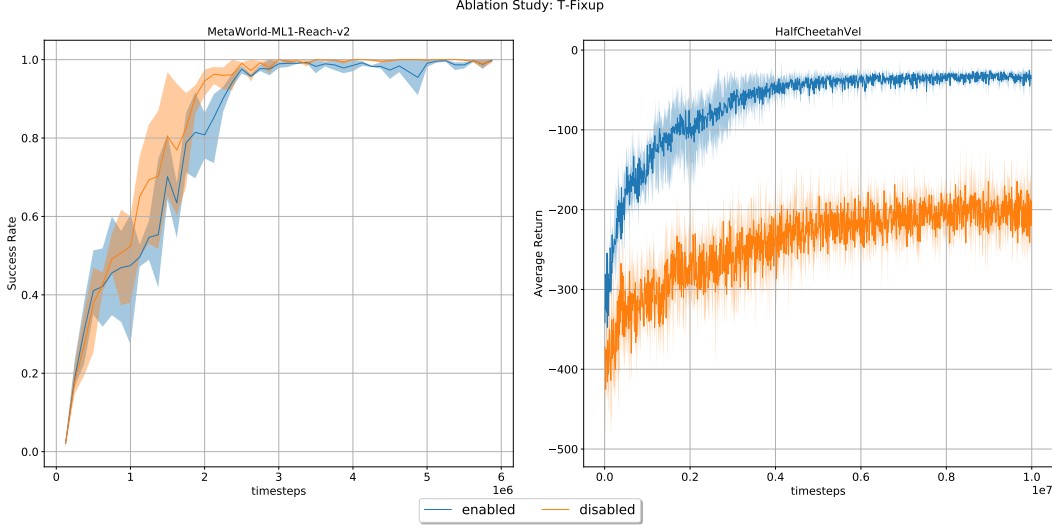

Figure 10: Ablation results for the T-Fixup component.

## D.2   WORKING MEMORY SEQUENCE LENGTH

A meta-RL agent requires a sequence of interactions to identify the running task and act accordingly. The length of this sequence $N$ should be large enough to address the ambiguity associated with the set of tasks, but not too long to make the transformer optimization harder and less sample efficient. In this ablation, we study two environments that present different levels of ambiguity and show that they also require different lengths to achieve optimal sample efficiency.

We first analyze MetaWorld-ML1-Reach-v2. The environment defines each target location in the 3D space as a task. The associated reward is the distance between the gripper and this target. Hence,

at each timestep, the reward is ambiguous for all the tasks located on the sphere's surface with the center in the gripper position. This suggests that the agent will benefit from long sequences. Figure 11 (left) confirms this hypothesis, as the sample efficiency improves until sequences with several timesteps (N = 50).

The HalfCheetahVel environment defines each forward velocity as a different task. The associated reward depends on the difference between the current cheetah velocity and this target. Hence, at each timestep, the emitted reward is ambiguous only for two possible tasks. To identify the current task, the agent needs to estimate its velocity (which requires a few timesteps) and then disambiguate between these two tasks. This suggests that the agent will not benefit from very long sequences. Figure 11 (right) confirms this hypothesis: there is an improvement from N = 1 to N = 5, but the performance decreases for longer sequences as the training becomes harder.

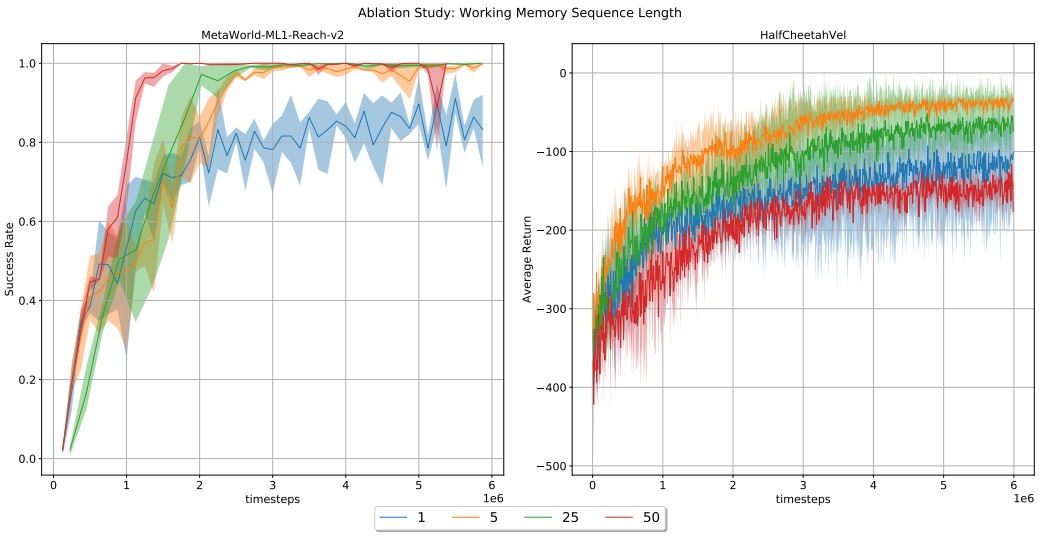

Figure 11: Ablation results for the working memory sequence length.

### D.3    NUMBER OF LAYERS

Another important component is the network depth. In Section 4, we hypothesized that more layers would help to recursively build a more meaningful version of the episodic memory since we interact with output memories from the past layer and mitigates the bias effect from the task representations. Figure 12 shows how TrMRL behaves according to the number of layers. We observe a similar pattern to the previous ablation case. For Reach-v2, more layers improved the performance by reducing the effect of ambiguity and biased task representations. For HalfCheetaVel, we can see an improvement from a single layer to 4 or 8 layers, but for 12 layers, sample efficiency starts to decrease. On the other hand, we highlight that even for a deep network with 12 layers, we have a stable optimization procedure, showing the effectiveness of the T-Fixup initialization.

### D.4    NUMBER OF ATTENTION HEADS

The last ablation case relates to the number of attention heads in each MHSA block. We hypothesized that multiples heads would diversify working memory representation and improve network expressivity. Nevertheless, Figure 13 shows that more heads slightly increased the performance in HalfCheetahVel and did not interfere in Reach-v2 significantly.

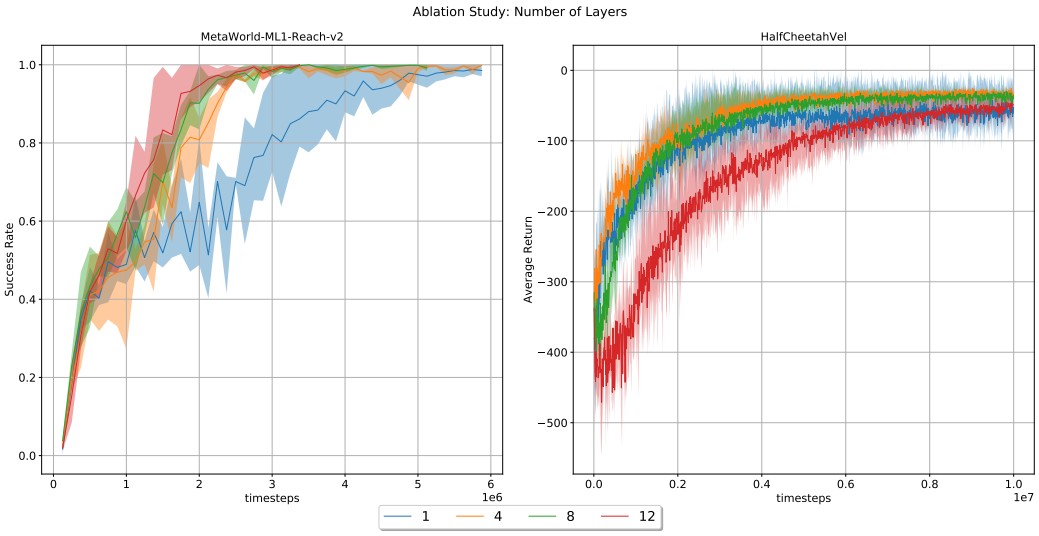

Figure 12: Ablation study for the number of transformer layers.

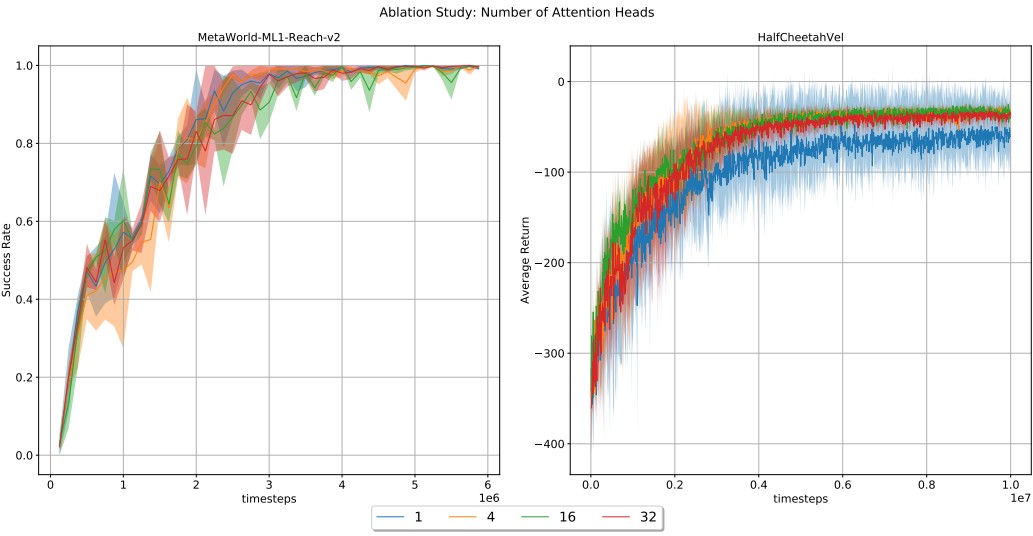

Figure 13: Ablation study for the number of attention heads.

# E    PROOF OF THEOREM 1

**Theorem 1.** *Let $\mathcal{S}^l = (e_0^l, \ldots, e_N^l) \sim p(e|\mathcal{S}^l, \boldsymbol{\theta}_l)$ be a set of normalized episodic memory representations sampled from the posterior distribution $p(e|\mathcal{S}^l, \boldsymbol{\theta}_l)$ induced by the transformer layer $l$, parameterized by $\boldsymbol{\theta_l}$. Let $K$, $Q$, $V$ be the Key, Query, and Value vector spaces in the self-attention mechanism. Then, the self-attention in the layer $l+1$ computes a consensus representation $e_N^{l+1} = \frac{\sum_{t=1}^N e_t^{l,V} \cdot \exp \langle e_t^{l,Q}, e_i^{l,K} \rangle}{\sum_{t=1}^N \exp \langle e_t^{l,Q}, e_i^{l,K} \rangle}$ whose associated Bayes risk (in terms of negative cosine similarity) lower bounds the Minimum Bayes Risk (MBR) predicted from the set of candidate samples $\mathcal{S}^l$ projected onto the $V$ space.*

*Proof.* Let us define $\mathcal{S}_V^l$ as the set containing the projection of the elements in $\mathcal{S}^l$ onto the $V$ space: $\mathcal{S}_V^l = (e_0^{l,V}, \ldots, e_N^{l,V})$, where $e^{l,V} = W_V \cdot e^l$ ($W_V$ is the projection matrix). The Bayes risk of selecting $\hat{e}^{l,V}$ as representation, $BR(\hat{e}^{l,V})$, under a loss function $\mathcal{L}$, is defined by:

$$BR(\hat{e}^{l,V}) = \mathbb{E}_{p(e|\mathcal{S}_V^l, \boldsymbol{\theta}_l)}[\mathcal{L}(e, \hat{e})] \tag{7}$$

The MBR predictor selects the episodic memory $\hat{e}^{l,V} \in \mathcal{S}_V^l$ that minimizes the Bayes Risk among the set of candidates: $e_{MBR}^{l,V} = \arg\min_{\hat{e} \in \mathcal{S}_V^l} BR(\hat{e})$. Employing negative cosine similarity as loss function, we can represent MBR prediction as:

$$e_{MBR}^{l,V} = \arg\max_{\hat{e} \in \mathcal{S}_V^l} \mathbb{E}_{p(e|\mathcal{S}_V^l, \boldsymbol{\theta}_l)}[\langle e, \hat{e} \rangle] \tag{8}$$

The memory representation outputted from a self-attention operation in layer $l+1$ is given by:

$$e_N^{l+1} = \frac{\sum_{t=1}^N e_i^{l,V} \cdot \exp \langle e_t^{l,Q}, e_i^{l,K} \rangle}{\sum_{t=1}^N \exp \langle e_t^{l,Q}, e_i^{l,K} \rangle} = \sum_{t=1}^N \alpha_{N,t} \cdot e_t^{l,V} \tag{9}$$

The attention weights $\alpha_{N,t}$ define a probability distribution over the samples in $\mathcal{S}_V^l$, which approximates the posterior distribution $p(e|\mathcal{S}_V^l, \boldsymbol{\theta}_l)$. Hence, we can represent Equation 9 as an expectation: $e_N^{l+1} = \mathbb{E}_{p(e|\mathcal{S}_V^l, \boldsymbol{\theta}_l)}[e]$. Finally, we compute the Bayer risk for it:

$$\begin{aligned}
BR(e_N^{l+1}) &= \mathbb{E}_{p(e|\mathcal{S}_V^l, \boldsymbol{\theta}_l)}[\langle e, \hat{e}_N^{l+1} \rangle] \\
&= \mathbb{E}_{p(e|\mathcal{S}_V^l, \boldsymbol{\theta}_l)}[\langle e, \mathbb{E}_{p(e|\mathcal{S}_V^l, \boldsymbol{\theta}_l)}[e] \rangle] \\
&= \left\langle \mathbb{E}_{p(e|\mathcal{S}_V^l, \boldsymbol{\theta}_l)}[e], \mathbb{E}_{p(e|\mathcal{S}_V^l, \boldsymbol{\theta}_l)}[e] \right\rangle \\
&\geq \left\langle \mathbb{E}_{p(e|\mathcal{S}_V^l, \boldsymbol{\theta}_l)}[\langle e, \hat{e} \rangle], \hat{e} \right\rangle \\
&= \mathbb{E}_{p(e|\mathcal{S}_V^l, \boldsymbol{\theta}_l)}[\langle e, \hat{e} \rangle], \forall \hat{e} \in \mathcal{S}_V^l.
\end{aligned} \tag{10}$$

□

# F  PSEUDOCODE

In this section, we present a pseudocode for TrMRL's agent during its interaction with an arbitrary MDP.

---

**Algorithm 1** TrMRL – Forward Pass

---

**Require:** MDP $\mathcal{M} \sim p(\mathcal{M})$
**Require:** Working Memory Sequence Length $N$
**Require:** Parameterized function $\phi(\boldsymbol{s}, \boldsymbol{a}, r, \eta)$
**Require:** Transformer network with $L$ layers $\{f_1, \ldots, f_L\}$
**Require:** Policy Head $\pi$
  Initialize Buffer with $N - 1$ PAD transitions: $\mathcal{B} = \{(\boldsymbol{s}_{PAD}, \boldsymbol{a}_{PAD}, r_{PAD}, \eta_{PAD})_i)\}, i \in \{1, \ldots, N - 1\}$
  $t \leftarrow 0$
  $\boldsymbol{s}_{next} \leftarrow \boldsymbol{s}_0$
  **while** episode not done **do**
      Retrieve the $N - 1$ most recent transitions $(\boldsymbol{s}, \boldsymbol{a}, r, \eta)$ from $\mathcal{B}$ to create the ordered subset $\mathcal{D}$
      $\mathcal{D} \leftarrow \mathcal{D} \bigcup (\boldsymbol{s}_{next}, \boldsymbol{a}_{PAD}, r_{PAD}, \eta_{PAD})$
      Compute working memories:
          $\phi_i = \phi(\boldsymbol{s}_i, \boldsymbol{a}_i, r_i, \eta_i), \forall \{\boldsymbol{s}_i, \boldsymbol{a}_i, r_i, \eta_i\} \in \mathcal{D}$
      Set $e_1^0, \ldots, e_N^0 \leftarrow \phi_1, \ldots, \phi_N$
      **for each** $l \in 1, \ldots, L$ **do**
          Refine episodic memories:
              $e_1^l, \ldots, e_N^l \leftarrow f_l(e_1^{l-1}, \ldots, e_N^{l-1})$
      **end for**
      Sample $\boldsymbol{a}_t \sim \pi(\cdot | e_N^L)$
      Collect $(\boldsymbol{s}_{t+1}, r_t, \eta_t)$ interacting with $\mathcal{M}$ applying action $\boldsymbol{a}_t$
      $\boldsymbol{s}_{next} \leftarrow \boldsymbol{s}_{t+1}$
      $\mathcal{B} \leftarrow \mathcal{B} \bigcup (\boldsymbol{s}_t, \boldsymbol{a}_t, r_t, \eta_t)$
      $t \leftarrow t + 1$
  **end while**

---

