# OpenReview forum: "Transformers are Meta-Reinforcement Learners"
_ICLR.cc/2022/Conference — ICLR 2022 Submitted_

### Official Review · Reviewer_ouZ3 · 2021-10-31

**Correctness:** 2
**Technical Novelty And Significance:** 1
**Empirical Novelty And Significance:** 1
**Recommendation:** 3
**Confidence:** 5

**Main Review:**

Strengths:
1) The paper attempts to study an alternative network architecture (i.e. transformer) in RL, which is a direction that has a lot of importance and potential.
2) The authors provide an explanation on how transformers can be a natural fit for memory-based meta RL. They argue that the attention mechanism can straightforwardly implement memory reinstatement.

Weaknesses:
1) Re the transformer use: TrMRL proposes to use transformers exactly as if they were RNNs in RL^2. The authors suggest to use the current transitions' embedding to input to the policy. This is a trivial extension of the prior work and a very obvious one. Moreover it has been done in various prior works and in several different settings. This basically makes TrMRL to be a version of RL^2 where an RNN is replaced with a transformer.
2) Re PEARL results: the demonstrated learning curves of PEARL directly contradict the claims from the original paper and my intuition. I think that it is very unlikely that TRPO based MAML or PPO based RL^2 (both are on-policy) can outperform that dramatically SAC-based PEARL (off-policy). It is a common knowledge that off-policy methods are more sample efficient than the on-policy methods. This also makes me question the correctness of the whole experiment, as TrMRL also uses PPO as an RL backbone.
3) Re T-Fixup initialization: the claim that the T-Fixup initialization helps is not supported in any experiment or an ablation study. It thus unclear how useful it is or essential.
4) Re Fig 4: Why does this compare performance during meta-training rather than meta-testing? In the end we are more interested in the latter.
5) Re Fig. 5: It would be more informative if other baselines are plotted here to server as a reference point.


**Summary Of The Paper:**

This work studies the use of the transformer architecture to enable memory-based meta RL. The authors argue that transformer's attention is  inherently suitable to facilitate episodic memory which can be then inputed to the policy. Furthermore, the authors point out that transformer demonstrates better properties at  handling longer transition sequences. Finally, the authors suggest that training a transformer in RL is often unstable, and to this end they propose to apply T-Fixup initialization that evidently enables stable training.
The paper provides an empirical study on a several high-dimensional continuous control tasks that feature locomotion and dexterous manipulation to support the aforementioned claims.

**Summary Of The Review:**

While that paper studies an important direction in RL of adopting new network architectures, such as transformers, the paper, unfortunately, brings very little novelty or insight.  The proposed method largely builds off of the existing work (such as RL^2) and essentially proposes to replace RNNs with transformers. The empirical study is also questionable and,  in my opinion, misleading. Moreover, the experimental methodology is a suspect as it doesn't provide an extensive ablation or support all the claims.

Taking this into account I suggest to reject the paper, as it in its current form doesn't meet the acceptance bar.

---

> ### Author Response · Authors · 2021-11-23
> **Some clarifications and details about the incorporated feedback**
>
> Dear reviewer ouZ3,
>
> Thanks for the review! We really appreciate the way the reviewer summarized our paper and pointed out the pros of our work. We also would like to clarify some concerns pointed out:
>
> *Concern (1)*
>
> Transformer use: We understand your point of view as employing transformers in the meta-RL setting directly sounds methodologically straightforward. Indeed, the optimization procedure presented by RL^2 is commonly used in prior work for network-based meta-learners [1, 2], where the “outer loop” is an on-policy model-free RL algorithm and the “inner loop” is learned by network weights.
> On the other side, we argue that our contribution relies on stabilizing the transformer network trained in the meta-RL objective, as this is a hard challenge empirically speaking. As an illustration, prior work [1] tried to employ this technique for the meta-RL setting and reported random performance. In this sense, we “re-opened” this door of transformers as meta-reinforcement learners for further research.
> Not only that: we showed the effectiveness of this method: TrMRL outperformed the on-policy baselines by a considerable margin in all scenarios evaluated.
>
> We are also very happy to discuss the prior work and different settings mentioned by the author during the rebuttal period. To the best of our knowledge, our work is the first one to effectively show transformers as meta-reinforcement learners. Additionally, speaking on the broader area of RL, this is also the first one that stabilizes transformers without demonstration/expert data, architectural additions, or restrictions in the observation space.
>
> *Concern (2)*:
>
> PEARL results: We would like to clarify that we strictly reproduced the results reported in the MetaWorld paper [3] for PEARL, MAML, and RL2 baselines. Indeed, we used the same implementation (provided by the Garage repository [4]), and the results are very similar. Note that we reported the first 50M timesteps in our work.
>
> We agree that the results sound counterintuitive at first glance, but we highlight that the claims from the PEARL paper are related to the evaluation in the MuJoCo benchmark, which is only composed of locomotion tasks and not dexterous manipulation tasks. Indeed, the PEARL results reported in our work (Appendix B) for the MuJoCo benchmark are aligned to the results from the original paper. Yu et al [3] hypothesized that PEARL’s poor performance in MetaWorld is related to the difficulty to train the task variational encoder in the MetaWorld tasks. We reinforce this since we also observed that MetaWorld tasks are more ambiguous to identify and harder to explore. In another perspective, prior work already pointed out the limited generalization capability from PEARL’s task encoder [5] and we also presented some evidence in this direction for OOD generalization (Figure 7) even for the MuJoCo benchmark.
>
> Given that the reviewer raised concerns about the correctness of the experiments, we also added a Reproducibility Statement section in our paper to further describe our reproducibility efforts.
>
> *Concern (3)*:
>
> T-Fixup Ablation Study: We would like to bring attention to the experiments in Appendix D for our ablations, and D.1 specifically for T-Fixup ablations. We presented how it is useful for environments where stabilizing learning from the beginning is very crucial to avoid poor policies.
>
> *Concern (4)*:
>
> Fig. 4: Here, we also strictly followed the evaluation protocol reported in the MetaWorld paper (see Figures 12-18 in there). Besides that, we believe that reporting performance on meta-training tasks could help understand multi-task learning properties (as part of the meta-RL process), while meta-testing tasks help understand adaptation/generalization properties. This requirement becomes more evident for ML45.
>
> *Concern (5)*:
>
> Fig. 5: Our idea was to only show TrMRL effectiveness on timestep-level online adaptation. We couldn’t plot for PEARL and MAML because they do not have such capability (they only present episode-level adaptation, requiring at least a few hundred timesteps to adapt).
>
>
> *Final Message*
>
> We are very open to discussing our work and the connections to the prior work, as we believe that TrMRL provides interesting empirical novelty and understanding. We also would like to highlight the supplementary evaluation in Appendix D, where we present ablations on the transformer structure to support our main claims.
>
> *References*
>
> [1] Mishra et al.A Simple Neural Attentive Meta-Learner. In: ICLR 2018.
>
> [2] Luisa Zintgraf et. al. VariBAD: A very good method for bayes-adaptive deep rl via meta-learning. In: ICLR, 2020.
>
> [3] Yu et al. A Benchmark and Evaluation for Multi-Task and Meta Reinforcement Learning. In: CoRL, 2020.
>
> [4] The garage contributors. Garage: A toolkit for reproducible reinforcement learning research. Available in:https://github.com/rlworkgroup/garage.
>
> [5] Russel Mendonca et al. Guided Meta-Policy Search. In: NeurIPS, 2019.

---

### Official Review · Reviewer_maih · 2021-11-02

**Correctness:** 2
**Technical Novelty And Significance:** 2
**Empirical Novelty And Significance:** 2
**Recommendation:** 3
**Confidence:** 3

**Main Review:**

This paper investigates whether the few-shot learning capabilities of Transformers make them well-suited to the type of task adaptation required in meta RL. This is an intriguing premise, and I agree that there is a connection here that is worth studying. Unfortunately, beyond the introductory premise I found this paper somewhat difficult to parse. Many components of the algorithm are stated qualitatively, without a precise enough description to be able to reproduce, such as:

> In this way, the transformer architecture recursively refines the episodic memory interacting output memories from the past layer.
As we compute each output memory in the perspective of a different timestep t, deeper layers reach
a consensus across the whole sub-trajectory.

I generally had a hard time understanding these descriptions in the paper. For this one in particular:
1. I am not sure what the recursive refinement refers to; could you write the recursion?
2. "Consensus" is not defined or used outside of this paragraph.
3. I think the "working memory" refers to feature vectors of recent transitions in the same spirit as the PEARL inference network $\Psi( z \mid s, a, s', r)$, and "episodic memory" is a function of summed working memory, as in Definition 6: $e_t = f(\mu_t^0, \ldots, \mu_t^\omega)$. However, as far as I can tell, $e_t$ is not referenced after its definition, so I do not know how it is used in the full algorithm. I also am not entirely sure what function $f$ is referring to. I assumed the memory terminology is the same as that in Fortunato et al 2019, but it would be better if these were precisely defined and motivated in this paper to make it more self-contained.

I did my best to keep pedantic issues off of that list and focus on areas where I genuinely found clarity issues. (For example, "from the perspective of" struck me as a slightly strange anthropomorphism of Equation 5, but it is clear from context what that means, so is not as important.)

While there is some focus given to the memory representation, there seem to be a number of missing details about the architecture and algorithm more generally. Most seriously, I could not find an actual objective aside from the general RL objective in the preliminaries. Since the experiments are in continuous control, presumably there needed to be some modification to the architecture described in Vaswani et al 2017?

The final stated contribution of this paper is a study of an important initialization method called T-Fixup. This ends up being a little underwhelming, with the ablation showing that it matters in one task but not in another. The explanation given is:
> This result suggests that stabilizing learning since the beginning is very critical in the RL setting. This is often the
moment with higher exploration rates that prevent sub-optimal policies.

The first sentence seems a bit vacuously true, and I am not sure what the precise claim of the second is since the paper does not say anything about exploration.


**Summary Of The Paper:**

This paper presents a Transformer-based approach to the meta RL problem, in which self-attention serves as a memory lookup mechanism for task adaptation. Experimental evaluation focuses on continuous control meta RL benchmarks.

**Summary Of The Review:**

My main concerns about this paper are all related to clarity. It would help to have the full algorithm written precisely in one consolidated place (even if it's high-level pseudocode), rather than spread throughout the paper in informal descriptions with undefined terms.

I would be happy to revisit this if the authors would like to clear things up during the discussion period.

---

> ### Author Response · Authors · 2021-11-23
> **Some clarifications and details about the incorporated feedback**
>
> Dear Reviewer maih,
>
> Thank you very much for your review. We found it very helpful since it catches very critical points to understand our work. We worked on a new version to address all concerns raised by the review:
>
> We rewrote Section 4.3 to replace the vague ideas and anthropomorphism to give place to more concrete and objective explanations. In more details, we:
> - Added how the recursion works (including pseudocode, as recommended);
> - Mathematically defined “consensus” in terms of Bayes Risk;
> - Added the definition of episodic memory in neuroscientific terms and how we concretely implement it in our work.
> - Added the definition of working memory in neuroscientific terms, while our implementation for it is described in Section 4.1.
>
> We also added a Theorem that shows how the recursion refines the episodic memory via this consensus representation. The proof of this theorem is in Appendix E.
>
> Additionally, we rewrote the T-Fixup ablation study (Appendix D.1), to make our claims clearer and to show how the results support them.
>
> *In terms of the RL objective*:
>
> We trained the TrMRL agent using PPO. Therefore, the RL objective is the same as stated in [1]. In our perspective, this is surprisingly good, as we could stabilize a vanilla transformer model training that is solely based on RL gradients (which are in general very noisy) from vanilla PPO.
>
> *Modifications for continuous control*:
>
> While Vaswani et al. [2] add a softmax layer on top of the transformer network, TrMRL adds a gaussian MLP as commonly used for RL policies. This is illustrated in Figure 1.
>
> We hope these changes can address your concerns and we are very open to further discussing any details of our works. Thank you so much!

---

### Official Review · Reviewer_irGj · 2021-11-02

**Correctness:** 3
**Technical Novelty And Significance:** 2
**Empirical Novelty And Significance:** 2
**Recommendation:** 5
**Confidence:** 4

**Main Review:**

Strengths:
* The paper addresses an important topic of Transformer architecture for meta-RL
* The paper makes a valuable comparison of Transformer training settings and their respective impact on meta-learning
* The preliminary results on meta-world are rather encouraging.

Weaknesses:
* Several important baselines are missing like (https://proceedings.mlr.press/v139/dance21a.html and https://arxiv.org/abs/1801.01290 )
* The novelty of the model remains quite limited compared to the SoA Transformer current utilization in sequential decision making
* The experimental results don't seem significant in ML45 settings which is an important transfer setting of Meta-World
*  Maybe a connection to auto-prompting for zero-shot adaptation in controlled generation could have been interesting to develop (https://aclanthology.org/2020.emnlp-main.346.pdf).

**Summary Of The Paper:**

The paper is proposing TrMRL (Transformers for Meta-Reinforcement Learning), a memory-based meta-Reinforcement Learner which uses the transformer architecture to formulate the learning process.
It works as a memory associating recent working information to create an episodic memory which is used to contextualize the policy.
The paper deals with the recently discovered limitations of Transformer training that exhibit been important to consider the particular context of sequential decision making and over-task generalization given an underlying environment dynamics.
The approach is evaluated on Metaworld in various task transfer settings.

**Summary Of The Review:**

The paper is proposing an improvement of the transformer architecture, introducing a mechanism of memory reinstatement for meta-learning in sequential decision making.
The problem addressed in the paper is important and up-to-date.
Unfortunately, several state-of-the-art approaches are missing in the evaluation part.
Moreover, while several preliminary results are encouraging, the results depicted in ML45 remain limited compared to the proposed baseline approaches.

---

> ### Author Response · Authors · 2021-11-23
> **Some clarifications and details about the incorporated feedback**
>
> Dear Reviewer irGj,
>
> Thank you for your review. We incorporated your feedback and we also would like to clarify some points:
>
> *Concern (1):*
>
> From the methods mentioned as baselines:
>  - For SAC (https://arxiv.org/abs/1801.01290): PEARL is a meta-RL algorithm that employs the SAC framework as part of its training. SAC is a very important baseline for the broad RL setting, and PEARL adapts it to the meta-RL setting. Given this difference, the natural baseline choice for the paper setting (meta-RL) is PEARL rather than SAC.
>  - DCRL  (https://proceedings.mlr.press/v139/dance21a.html) brings a very interesting contribution to the Few-Shot Imitation setting, which is slightly different from the meta-RL setting considered here as we don’t have access to expert demonstrations or any previous interaction data. In the “classic” meta-RL setting applied here, the algorithm must explore from scratch to learn and adapt. Given the difference in the scenarios, we think that is reasonable not to include it in our evaluation.
>
> In our work, we considered a strong and diverse set of baselines to cover the main approaches for meta-RL in the recent literature: PEARL brings the sample efficiency from off-policy methods with variational inference for task identification; MAML brings the idea of adaptation via gradient descent in few samples; RL2 represents the memory-based meta-learner via RNNs. These are also the algorithms presented in the MetaWorld paper [1], as they are the key algorithms to the area.
>
> *Concern (2)*:
>
> Regarding the current utilization of transformers in sequential decision making:
>
> We are assuming here that the reviewer is referring to the Decision Transformers [2] or Trajectory Transformers [3]. Indeed, they are really interesting methods with breakthrough results in the context of offline RL/Imitation Learning. Again, this is different from our scenario here: they use demonstration data (including expert data) and casts the problem as a sequence prediction. This simplifies the problem a lot since these methods do not need to explore or apply RL gradients (they are not maximizing rewards), which is well-known to be much noisier [4]. TrMRL, on the other side, collects its own data (with exploration) and maximizes rewards from an RL objective.
>
> In our perspective, these methods and TrMRL are complimentary. If we consider the interesting and strong contributions from all of these methods, they can bring a lot of attention back to the research of transformers as end-to-end (meta)-Reinforcement Learners.
>
> *Concern (3)*:
>
> ML45 Results:
>
> We agree that none of the methods performed well in this benchmark. In fact, to the best of our knowledge, no methods could actually present good performance in ML45, which shows that there is a lot to improve in the meta-RL setting. We presented ML45 results to be very transparent regarding the limitations of TrMRL and SoA methods. It is worth mentioning that we are here strictly following the benchmark - some methods present better Success Rates because they use expert data or modify rewards to simplify the problem (this is the case of DCRL, for example).
>
> From a different perspective, TrMRL presented the best performance in terms of training tasks fitting. This suggests an effective multi-task learning approach, especially considering that ML45 has a very diverse set of tasks. We hypothesize here that adding a proper mechanism for meta-exploration on top of TrMRL will improve ML45 performance considerably, given the adaptability and generalization results from the other evaluation scenarios. We judged this as outside of the scope of the current work.
>
> *Concern (4)*:
>
> Connection with AutoPrompt: Thank you for the recommendation! We believe that this aligns well with future research on transformers for meta-RL. Indeed, one of the most effective ways to stabilize training and further improve sample efficiency is to rely on other forms of self-supervision that enable important model capabilities. One good example in RL is curiosity exploration via self-supervised prediction [5]. In this sense, AutoPrompt brings an interesting insight to augment tasks that will help for multi-task training and further adaptation. We added one paragraph about it in Section 6.
>
> Thanks for your review! We are happy to further discuss any other concerns.
>
> *References*
>
> [1] Yu et al. Meta-World: A Benchmark and Evaluation for Multi-Task and Meta Reinforcement Learning. Proceedings of the Conference on Robot Learning, 2020.
>
> [2]  Chen et al. Decision transformer: Reinforcement learning via sequence modeling, 2021.
>
> [3] Janner et al. Reinforcement learning as one big sequence modeling problem, 2021.
>
> [4] Norouzi et al. Reward augmented maximum likelihood for neural structured prediction, 2016.
>
> [5] Pathak et all. Curiosity-driven exploration by self-supervised prediction. In Proceedings of the 34th International Conference on Machine Learning, 2017.

---

### Official Review · Reviewer_VA5K · 2021-11-03

**Correctness:** 3
**Technical Novelty And Significance:** 3
**Empirical Novelty And Significance:** 3
**Recommendation:** 5
**Confidence:** 4

**Main Review:**

Pros:
(1) This paper introduces a new variant of meta-reinforcement learning by leveraging transformer, which can capture long sequence and context dependence for fast adaption in new tasks. And the intuition to leverage transformer makes sense.
(2) It is well written and easy to understand and follow. There are few typos here and there but nothing that affects the readability of the paper.
(3) They performed ablation study on how transformer architecture affects the performance.

Cons:
(1) In the idea level, it is trivial to combine transformer model with meta-reinforcement learning to learn hidden distribution over different tasks for fast adaption. As RL^2, it builds the policy from the episodic memory generated by the transformer, instead of RNN.
(2) The experimental results are not convincing. If I do not misunderstand it, I think N is the sequence length (correct me if it is wrong). The authors claim in the abstract that transformer can capture long sequence and context dependence for fast adaption, but why we set N=2 for all experiments? Is the transformer a good fit here? The experiment in appendix D.2 show it cannot get better result with long sequence. In Fig. 8, I also see it is not comparable to PEARL.

Minor comments:
(1)	A typo in Eq. 5? Is it a summation from n=1 to t?
(2)	N=2 for all experiments? Can you try different N in Fig. 8, in order to compare with PEARL?



**Summary Of The Paper:**

This paper presents TrMRL (Transformers for Meta-Reinforcement Learning), which leverages transformer to learn hidden distributions for meta-reinforcement learning. Specifically, the authors claim that the model associates the recent past of working memories to build an episodic memory recursively through the transformer layers, which can help to adapt to new tasks. The conducted experiments in high-dimensional continuous control environments for locomotion and dexterous manipulation show that TrMRL achieves comparable results on performance, sample efficiency and out-of-distribution generalization in these environments.

**Summary Of The Review:**

This paper introduces a new variant of meta-reinforcement learning by leveraging transformer, which can capture long sequence and context dependence for fast adaption in new tasks. Empirically, the proposed method shows some improvements over the existing ones. However, a few major concerns are as follows.
(1) The authors claim in the abstract that transformer can capture long sequence and context dependence for fast adaption, but why do we set N=2 for all experiments? Is the transformer a good fit here? The experiment in appendix D.2 show it cannot get better result with long sequence.
(2) In Fig. 8, its performance is not comparable to PEARL. Is it possible to try different N and compare with PEARL?

Overall, it is a good paper, but not good enough for publication. More convincing experiments needed to show the advantage of transformer here.

---

> ### Author Response · Authors · 2021-11-23
> **Some clarifications and details about the incorporated feedback**
>
> Dear Reviewer VA5K,
>
> Firstly, thank you very much for your review! You raised very important points about the paper. Hopefully, we can clarify some of the concerns:
>
> *Concern (1)*:
>
> We agree that reframing the meta-RL procedure as a transformer learning problem sounds straightforward methodologically since that the transformer architecture and memory-based meta-learners (such as RL2) are well known and widely used in recent literature. Nevertheless, the key point is that stabilizing a transformer network trained in the meta-RL objective is a hard and open challenge (which our work addresses). In this line, we argue that TrMRL presents an interesting empirical contribution to the field.
>
> Indeed, transformers are much harder to train than other architectures (such as RNNs), requiring many tricks even in supervised learning. Prior work [1] tried to employ this technique for the meta-RL setting and reported random performance. Here, we reported gains in sample efficiency or post-adaptation performance for many scenarios (or at least comparable evaluation) over a strong and diverse set of baselines. Even in the broader scope of RL, the works that employed transformers end-to-end required architectural additions [2], imposed restrictions to observations [3], or re-casted the training as a supervised learning problem [4, 5]. None of them were able to apply the vanilla transformer with the RL objective (i.e., maximizing rewards) without constraints to the MDPs. TrMRL avoids all these requirements/restrictions by solely applying a different weight initialization method (T-Fixup), which is an interesting finding for the RL field in our perspective.
>
>
> *Concern (2)*:
>
> Setting N = 2 for all experiments: We would like to apologize for our mistake here. We used the N symbol with two different meanings in the text. In Equation (3), it represents the sequence length that we feed in TrMRL. In the section 5.1, on the other side, it represents the number of episodes concatenated during PPO optimization (which is indeed set equals to 2). We fixed this mistake and introduced “E” (instead of N) for the second scenario.
>
> In terms of sequence length that we feed in TrMRL, we varied depending on the experiment, with a maximum of 50 timesteps. We rewrote the ablation discussion to better support our claims and validate the need for long sequences for certain environments. To summarize, our understanding is that the sequence length required is intrinsically related to the uncertainty/ambiguity associated with the task identification.
>
> *Concern (3)*:
>
> Comparability with PEARL in locomotion tasks: Yes, PEARL achieved better performance than TrMRL in the scenario of Appendix B. PEARL is a very strong baseline for locomotion tasks due to its off-policy nature inherited from the SAC framework. It is hard for TrMRL to achieve the same sample efficiency here since it is based on PPO (which is on-policy and well known to be more sample hungry). Nevertheless, TrMRL presents other advantages over PEARL:
>
> - TrMRL performed consistently in both locomotion and dexterous manipulation tasks. PEARL is very well suited for locomotion tasks but failed to learn dexterous manipulation. Prior work already points out the difficulty to train PEARL’s task encoder for dexterous manipulation [6];
> - Even in locomotion tasks, PEARL fails to generalize for OOD tasks (see Figure 7) since it cannot generate useful latent representations [7]. TrMRL reported the best performance in this scenario. It is worth mentioning that we use the exact same agents in the experiments for Figures 7 and 8.
> - By design, PEARL does not support online adaptation (i.e., adaptation at the timestep level). TrMRL does present (Figure 5).
>
> We hope that the aforementioned points can address the reviewer’s concerns. We also welcome further discussion/clarifications.
>
> *References*
>
> [1] Mishra et al. A Simple Neural Attentive Meta-Learner. In: International Conference on Learning Representations, 2018.
>
> [2] Emilio Parisotto et al. Stabilizing transformers for reinforcement learning. In Proceedings of the 37th International Conference on Machine Learning, 2020.
>
> [3] Loynd et al. Working memory graphs. Proceedings of the 37th International Conference on Machine Learning, 2020.
>
> [4]  Lili Chen et al. Decision transformer: Reinforcement learning via sequence modeling, 2021.
>
> [5] Janner et al. Reinforcement learning as one big sequence modeling problem, 2021.
>
> [6] Yu et al. Meta-World: A Benchmark and Evaluation for Multi-Task and Meta Reinforcement Learning. Proceedings of the Conference on Robot Learning, 2020.
>
> [7] Russel Mendonca et al. Guided Meta-Policy Search. In Advances in Neural Information Processing Systems, 2019.

---

### Author Response · Authors · 2021-11-23
**Improved version of the paper**

Dear reviewers:

We really appreciated the detailed reviews and all the concerns raised. We worked to incorporate all the actionable feedback in our paper. Here, we present a list of changes:

- We added new theoretical insights for TrMRL: we introduced a mathematical framework to define consensus representation in terms of Bayes Risk and how it works to refine episodic memory representation (Theorem 1). We also added more concrete definitions for working and episodic memories. We describe their implementation details in Sections 4.1 and 4.3, respectively.

- We added a Reproducibility Statement (Section 7) to describe all our efforts to make the results reproducible, for both the proposed method and the baselines. We hope to address any concerns in terms of our experimental methodology.

- We rewrote our analysis for the ablations regarding T-Fixup (Appendix D.1) and Sequence Length (Appendix D.2) to improve the clarity of our claims and how the experiments support it.

- We improved Section 6 to include longer-term future work, involving self-supervision as a research line to further improve sample efficiency in the meta-RL setting.

---

### Decision · Program_Chairs · 2022-01-20

**Decision:**

Reject

**Comment:**

At a high level, the novelty of this paper is limited: RL2 with transformers instead of RNNs. The emphasis is then placed on the experimental evaluation. Unfortunately, the reviewers felt that the experimental methodology and results were not strong enough at this stage to warrant publication. During the rebuttal, the reviewers did not engage nor discuss the author response, unfortunately, so I do not know what they think of the rebuttal. However, on evaluating the concerns of the reviewers against the updated manuscript, I think the updates do not go far enough to satisfy the concerns raised (experiments + baselines). Therefore, I recommend rejection.